# Single-item versus scale: Comparing respondent demographic, social, and health characteristics by measure of loneliness using the Canadian Longitudinal Study on Aging (CLSA) data

Stephanie A. Chamberlain[1]*, Lauren E. Griffith[2], Rachel D. Savage[3,4,5], Jesse Batara[6], Wenting Yan[1], Andrea Gruneir[3,4,6]

**1** Faculty of Nursing, College of Health Sciences, University of Alberta, Edmonton, Alberta, Canada, **2** Department of Health Research Methods, Evidence, and Impact, Faculty of Health Sciences, McMaster University, Hamilton, Ontario, Canada, **3** Women's College Research Institute, Women's College Hospital, Toronto, Ontario, Canada, **4** ICES, Toronto, Ontario, Canada, **5** Institute of Health Policy, Management and Evaluation, Dalla Lana School of Public Health, University of Toronto, Toronto, Ontario, Canada, **6** Department of Family Medicine, Faculty of Medicine and Dentistry, College of Health Sciences, University of Alberta, Edmonton, Alberta, Canada

* sachambe@ualberta.ca

## Abstract

### Background

Growing research and clinical attention to loneliness has led to a wide range of available measures; however, few studies directly compare different loneliness measures within the same population. Measures include single-item, direct questions and indirect, multi-item scales. This study compared responses and respondent characteristics between a direct single-item loneliness question and an indirect multi-item scale.

### Methods

We conducted a cross-sectional analysis of population-based survey data from the Canadian Longitudinal Study on Aging (CLSA). The sample included baseline (2010−2015) and follow-up 1 respondents (2015−2018) from both the Tracking (computer-assisted telephone interviews) and Comprehensive (in-home assessment) cohorts (n = 43,235). Loneliness was assessed using a direct single-item question from the Center for Epidemiological Studies Depression Scale (CES-D-10) and the 3-item Loneliness Scale. We categorized loneliness using ordinal (not lonely, moderately lonely, severely lonely) and dichotomous (lonely/not lonely) classifications. Unweighted descriptive statistics were used to examine demographic, social, and health characteristics across measures and response categories.

**Data availability statement:** Data are available from the Canadian Longitudinal Study on Aging (www.clsa-elcv.ca) for researchers who meet the criteria for access to de-identified CLSA data.

**Funding:** This work was supported by a Canadian Institutes of Health Research Catalyst Grant: Analysis of Canadian Longitudinal Study on Aging (CLSA) to AG (Grant #: 170313). The funder had not role in the design, conduct, interpretation, or writing of the manuscript. The opinions expressed in the manuscript are the author's own and do not reflect the views of the Canadian Longitudinal Study on Aging.

**Competing interests:** We have no competing interests to disclose.

**Abbreviations:** CLSA: Canadian Longitudinal Study on Aging; CCHS: Canadian Community Health Survey

## Results

Among the 43,235 CLSA respondents, 14% (n = 5,848) were identified by both the single-item question and the scale as lonely, 67% (n = 28,998) were identified as not lonely by both, leaving 19% classified differently based on measure. Individuals identified as lonely respondents by both measures were more often older, women, less educated, and had lower income. Those classified as severely lonely on the single item but not lonely on the scale were older than other groups. In contrast, respondents who were moderately lonely on the single-item question tended to be married, report higher income, and have greater social contact.

## Conclusion

The direct single-item question and the indirect multi-item scale provided comparable overall classifications of loneliness in our CLSA sample. However, respondent characteristics varied across the more granular response categories, highlighting the importance of measure selection depending on study aims.

## Background

Loneliness is associated with numerous negative health outcomes and increased risk of mortality [1–3]. It is prevalent in most developed countries [4], and due to its public health implications has generated substantial research activity [5,6]. Systematic reviews of loneliness have found over 60 measurement scales [7]; however, there is no consensus on "gold standard" measures and studies may vary on or recommendations for which cut-offs they use when classifying loneliness. Lack of guidance on the selection and the use of these measures creates challenges for research, surveillance, and screening, both from a comparability perspective and for clinicians and service providers trying to choose the best available measure for their setting.

Loneliness measures include multi-item scales and single-item questions. Measures range in the number of items, dimensionality (i.e., unidimensional view of loneliness versus multidimensional), mode of administration (e.g., telephone versus in-person), and degree of subjectivity (i.e., direct or indirect measure) [8–10]. Scales, such as the UCLA Loneliness Scale, the De Jong Gierveld Loneliness Scale, are considered indirect measures because they ask multiple questions related to the concept of loneliness without directly using the word lonely [11,12]. One of the most widely used scales to assess loneliness is the 3-item Loneliness Scale, a version of the Revised UCLA Loneliness Scale [11,13]. It is used in several national surveys including the Canadian Longitudinal Study on Aging (CLSA), the Canadian Community Health Survey (CCHS), and the Health and Retirement Survey in the United States [14–17]. This shorter version was developed because the original revised 20-item UCLA scale is self-administered, lengthy, and less feasible for large population studies, particularly those that rely on telephone interviews [11]. The 3-item Loneliness Scale is one of the few loneliness scales that has been validated for

telephone administration, making it particularly useful for large population-based surveys, and it is quick to complete. A psychometric comparison of the 3- and 20-item UCLA Loneliness Scales also supports the use of the 3-item scale in time- and resource-limited studies, as it shows comparable psychometric performance in terms of reliability and construct validity [18]. In addition, indirect measures like the 3-item Loneliness Scale are often preferred for respondent groups, such as older men, who are reluctant to self-identify as lonely [19]. Unlike indirect measures, single-item, direct measures specifically use the word lonely or loneliness. Direct measures are more controversial than indirect because of concerns that they elicit less truthful, more socially desirable responses because of the stigma associated with loneliness [10,20]. However, given the interest in screening for loneliness, single-item measures may prove useful in contexts where time and resources are limited such as primary care. A recent systematic review and meta-analysis examined the prevalence of loneliness across 113 countries and found that single-item direct measures were the most used assessment tools for loneliness [4].

Given the large number of different loneliness measures (both scales and single item), there have been few opportunities to compare multiple measures in the same respondents, even though the Office for National Statistics in the United Kingdom recommends using both types of capture a more complete picture of loneliness [19]. The CLSA includes the 3-item Loneliness Scale and a direct single-item question. Both scales assess loneliness by asking "how often do you feel," using a frequency-based scaling reference. However, to date, the CLSA does not offer recommendations for users on which measure to use, and responses to each measure have not been compared. This paper aims to address this gap by descriptively comparing responses to the direct (single item question) and indirect (3-item Loneliness Scale) measures of loneliness used in the CLSA. The purpose of this project was to describe the responses to the direct and indirect measure of loneliness in the CLSA and explore how these measures compare in relation to respondents' demographic, social, and health characteristics.

## Methods

We used a cross-sectional study design to descriptively analyse population-based survey data from the CLSA [17]. We examined data from baseline (2010–2015) and the first follow-up wave of survey data collection (2015–2018), when the 3-item Loneliness Scale was first included. The CLSA consists of two cohorts that are both age- and sex-stratified random samples of community-dwelling Canadians aged 45–85 years at recruitment. The random samples were stratified at the provincial level. The Tracking Cohort was recruited from across all 10 provinces with no geographic restrictions and completed a computer-assisted telephone interview. The Comprehensive Cohort includes individuals who live within 25–50 km of a CLSA data collection site and took part in in-home interviews and provided physical measures and biological data [21]. The CLSA exclusion criteria is described elsewhere [17].

### Study sample

Our sample (n = 44,816) included participants who completed both the baseline (2010–2015) and the first follow-up survey (2015–2018) of the CLSA. It consisted of 17,051 participants in the Tracking cohort and 27,765 participants in the Comprehensive cohort. Loneliness was measured in the Loneliness Scale only at follow-up 1, therefore no longitudinal comparisons or change could be examined. We excluded 1,581 respondents who did not answer both the 3-item Loneliness Scale and the single-item loneliness question or had other missing responses. This resulted in 3.53% (1,581/44,816) missing data. For the purposes of this descriptive study, we elected not to impute the missing data and used listwise deletion. The final sample was n = 43,235.

### Indirect measure: 3-item loneliness scale

The CLSA includes the 3-item Loneliness Scale [19]; one of most used scales to assess loneliness (S1 Table). It was first included in the first follow-up of the CLSA. It assesses loneliness using three separate questions: "How often do you feel left out?", "How often do you feel isolated from others?", and "How often do you feel that you lack companionship?". There

are three possible response categories: "1: Hardly ever", "2: Some of the time" and "3: Often". Scores range from 3 to 9, with higher scores indicating greater perceived loneliness. Exploratory and confirmatory factor analyses have demonstrated robust reliability and concurrent and discriminant validity when used in large, population-based samples [11].

### Direct measure: Single-item

The direct single-item question is embedded in the Center for Epidemiological Studies Depression Scale (CES-D-10). The question asks respondents to think about how they have felt in the past week, and if during this time, "How often do you feel lonely?" The overall score ranges from 0 (Rarely) to 3 (All the time), with higher scores indicating greater perceived loneliness.

### Other variables

We compared respondent demographic, health, and social factors for the single-item and scale measures. For demographics, we examined age, gender, education, income, marital status, and living arrangement (living alone). We examined the respondent's current gender (as opposed to biological sex), and classified respondents as one of: women (includes transwomen), men (includes transmen), and gender diverse (includes gender queer, other, do not know). Our other published work includes more information about our examination of the sex and gender variables in the CLSA [14,22]. We initially included variables measuring ethnicity, language, and geographic location. However, due to small proportions of our total sample (e.g., less than 3% were from non-White backgrounds), we did not report these variables. To assess health, we identified respondents' number of chronic conditions, functional impairment, number of depressive symptoms, and diagnosis of anxiety. We also examined self-reported health service use in the previous 12 months including emergency department visit, professional and non-professional care received, and perception of any unmet health care needs. Rather than using a composite index of social isolation, we examined individual social components, including living alone, martial status, social participation, and social contact to provide a descriptive assessment of factors associated with loneliness. To assess social factors, we examined number of social contacts and participation in social activities. Social contacts were identified by asking whether respondents had seen any of the following groups in the past six months: children, siblings, other relatives, close friends, or neighbours. Based on the number of contacts, respondents were categorized as having high (4–5 contacts), moderate (2–3 contacts), or low (0–1 contacts) levels of social contact. Social activity participation was identified using the same categorization based on the number of activities and was defined as whether respondents had taken part in any of the following within the last six months: family or friendship activities outside the household, church or religious activities, sports or physical activities, educational or cultural activities, or service club or fraternal organisation activities [14,22].

### Analysis

Using unweighted survey data, we calculated descriptive statistics (e.g., mean, standard deviation, frequencies, percentages) to compare demographic and health factors across categories of loneliness. To examine the degree of perceived loneliness, we used approaches that are consistent with the extant loneliness literature [2,4,23,24]. In general, quantitative measures aim to classify respondents based on their severity of loneliness ranging from not lonely to severely lonely

For the 3-item scale, to create an ordinal measure we used categories consistent with comparable studies and categorized participants as not lonely (score = 3), moderately lonely (score = 4 or 5), severely lonely (score ≥6) [23,24]. We noted that under this categorization, respondents who selected "some of the time" on all three items (total score = 6) were classified as severely lonely. To improve the robustness of our findings, we also conducted additional analysis for the 3-item scale, categorizing participants as not lonely (score = 3), moderately lonely (score = 4–6), severely lonely (score>7). We did not observe any significant differences based on changing the cut-off, so we elected to use the first approach (score >6

severely lonely) because this is consistent with other research in large, population-based surveys, as well as other studies that have analyzed the CLSA [14,22].

For the single item, we categorized participants as not lonely (Rarely or never, score = 1), moderately lonely (some of the time/occasionally, score = 2), or severely lonely (all of the time, score = 3). We also created a dichotomous measure (lonely/not lonely) and we identified individuals in the top quintile of our sample for both the scale and single item measure [2].

We calculated the correlation between the unweighted measures using Goodman-Kruskal's gamma (λ), Cohen's kappa (κ) to measure agreement [25]. We generated a series of contingency tables with frequencies and percents to identify concordant and discordant respondents. Once the contingency tables were completed for each measure (dichotomous, ordinal), we isolated the concordant and discordant respondents and assessed their demographic and health factors using descriptive statistics (frequency, percent, mean, standard deviation). To evaluate the consistency and potential differences in loneliness classification between the single-item and scale we applied the McNemar-Bowker test which is appropriate for contingency tables with more than two categories (e.g., not lonely, moderately lonely, severely lonely) and tests whether changes in responses are symmetric. A significant result indicates that the proportion of individuals reclassified between the categories differs across measures. Statistical software used in the analyses were IBM SPSS version 26 and SAS 9.4. We received research ethics approval from the University of Alberta (Pro00100416). CLSA participants provided written consent before participating in the study. All CLSA data were de-identified and authors had no access to identifiable information during the study period. Data for this project were accessed 01/03/2024.

## Results

The analytic sample included 43,384 respondents for the single-item loneliness measure with a mean age of 65.9 (SD 10.2) and 44,373 respondents for the 3-item scale with a mean age of 66.0 (SD 10.2). About half of the respondents (49%) were women, and more than half had less than a university education. Most respondents were identified as White, and lived in urban areas. Nearly one-quarter lived alone, and around one-fifth had four or more chronic conditions.

We found the single-item and scale had a moderate-strong positive correlation (λ = .80, p < 0.001). The prevalence of loneliness was similar for the two measures when dichotomized: 24% (n = 10,205) for the single-item measure versus 23% (n = 10,246) for the 3-item scale (S2 Table). For the ordinal measure, the prevalence of severe loneliness was also comparable: 11% (n = 4,618) on the single-item measure versus 12% (n = 5,463) on the 3-item scale. However there was variation in the number identified as moderately lonely, with 15% (n = 5,587) on the single-item measure versus 29% (n = 12,717) on the 3-item scale (S3 Table and S4 Table).

We found that 14% (n = 5,848) of the total sample were identified by both measures as lonely and 67% (n = 28,998) by both as not lonely, and 19% had a different classification across the two measures (Table 1). When stratified by gender, women followed a similar pattern to the total sample, with slightly more showing a different classification across the two measures (21%) compared to men (18%) (Table 1).

CLSA respondents who were identified as lonely by both measures differed from respondents identified as not lonely on both measures in several ways. Those identified as lonely (on both measures) were slightly older, more often women, less educated, and had lower income (Table 2). Compared to not lonely (on both measures) respondents, lonely respondents had different marital status distributions. A greater proportion of lonely respondents were widowed (21% of lonely respondents versus 7% of not-lonely respondents), divorced/separated (23% versus 9%), or single/never married (16% versus 6%), and there was a large discrepancy among those who were married (39% versus 78%). In terms of health, lonely respondents had a greater proportion of 4 + chronic conditions (33% versus 16%) and functional impairment (24% versus 10%). They had lower social contact (11% versus 6%) and low social participation (28% versus 15%). Compared to not-lonely, lonely respondents reported an unmet need in the previous 12 months (16% versus 6%) and having been to the

**Table 1. Comparing the Dichotomous (Lonely/Not Lonely) Responses from the 3-item Scale and Single-item Loneliness Measures for the Full Sample and Stratified by Gender (Women, Men).**

**Full sample (n = 43384)**

| Single item N (%) | 3-item Loneliness scale N (%) | | |
|---|---|---|---|
| | Lonely | Not Lonely | Total |
| Lonely | 5848 (14) | 4295 (10) | 10205 |
| Not Lonely | 4094 (9) | 28998 (67) | 33179 |
| Total* | 10246 | 34127 | 43384 |

**Women (n = 22094)**

| | 3-item Loneliness Scale N (%) | | |
|---|---|---|---|
| Single item N (%) | Lonely | Not lonely | Total |
| Lonely | 3384 (15) | 2381 (11) | 5765 |
| Not lonely | 2205 (10) | 14124 (64) | 16329 |
| Total* | 5589 | 16505 | 22094 |

**Men (n = 21091)**

| | 3-item Loneliness Scale N (%) | | |
|---|---|---|---|
| Single item N (%) | Lonely | Not lonely | Total |
| Lonely | 2453 (12) | 1907 (9) | 4360 |
| Not lonely | 1885 (9) | 14846 (70) | 16731 |
| Total* | 4338 | 16753 | 21091 |

*Percentages may not sum to 100 due to rounding

emergency department (29% versus 20%). Discordant respondents using the dichotomous measure of loneliness had a less obvious pattern of difference in their characteristics.

For each measure we categorized loneliness based on severity (not lonely, moderately lonely, severely lonely). We found fair agreement between the groups (ordinal measure weighted κ = 0.382, dichotomous measure weighted κ = 0.378) and found similar patterns emerge for the concordant respondents [25] (Table 3). Results of the McNemar-Bowker test found that for all groups there was significant (p < 0.001) asymmetry, indicating that individuals were re-classified differently across the two measures. The patterns of disagreement between the 3-item and single-item scales suggest that the two tools may not be interchangeable in identifying the severity of loneliness, as individuals were often classified as severely or moderately lonely based on the single-item cut-off, but were classified as not lonely by the 3-item scale. This misalignment was consistent across the full sample as well as in women and men, suggesting that the differences in classification are not driven by gender alone.

Compared to not lonely respondents, severely lonely respondents were slightly older (66.8 years versus 65.6), women, less educated, widowed, and living alone. Discordant respondents who were severely lonely in the single-item, direct measure but not lonely in the scale were older (69.5 years of age) than the other discordant and concordant groups (Table 4). Moderately lonely respondents (single-item) who were identified as not lonely in the scale, were married (67%), higher income (70%), high social contact (53%), and few unmet needs (8%) (S3 Table).

**Table 2. Descriptive characteristics of concordant and discordant respondents to scale and single-item loneliness measure.**

| | Concordant responses | | Discordant responses | |
|---|---|---|---|---|
| Variables | Responded that they were lonely (both single item and scale) | Responded that they were not lonely (both single item and scale) | Responded that they were lonely (single item) and not lonely (scale) | Responded that they were not lonely (single item) and lonely (scale) |
| | N (%) | N (%) | N (%) | N (%) |
| Total | 5848 (100) | 28998 (100) | 4295 (100) | 4094 (100) |
| **Age** | | | | |
| 46-55 | 992 (17) | 5289 (18) | 659 (15) | 778 (19) |
| 56-65 | 1982 (34) | 9748 (34) | 1305 (30) | 1395 (34) |
| 66-75 | 1518 (26) | 8475 (29) | 1177 (27) | 1100 (27) |
| 76-85 | 1143 (20) | 4842 (17) | 965 (22) | 694 (17) |
| 86+ | 213 (4) | 644 (2) | 189 (4) | 127 (3) |
| Age (Mean, SD) | 66.5 (10.5) | 65.6 (10.0) | 67.5 (10.6) | 65.6 (10.3) |
| **Gender** | | | | |
| Gender diverse | 9 (0.15) | 20 (0.06) | <6 | <6 |
| Women | 3384 (58) | 14124 (49) | 2381 (55) | 2205 (54) |
| Men | 2453(42) | 14846 (51) | 1907 (44) | 1885 (46) |
| **Education** | | | | |
| Less than a university degree | 3189 (55) | 14464 (50) | 2334 (54) | 2198 (54) |
| University degree or higher | 2142 (37) | 13100 (45) | 1639 (38) | 1607 (39) |
| **Income** | | | | |
| <$20,000 | 665 (11) | 786 (3) | 262 (6) | 274 (7) |
| $20,000-<$50,000 | 1898 (32) | 5409 (19) | 1199 (28) | 1095 (27) |
| $50,000+ | 2771 (47) | 21162 (73) | 2489 (58) | 2415 (59) |
| **Marital status** | | | | |
| Single, never married | 932 (16) | 1850 (6) | 460 (11) | 501 (12) |
| Divorced/separated | 1367 (23) | 2571 (9) | 661 (15) | 639 (16) |
| Married/common law | 2293 (39) | 22485 (78) | 2314 (54) | 2487 (61) |
| Widowed | 1255 (21) | 2078 (7) | 857 (20) | 464 (11) |
| **Living alone** | | | | |
| No | 2970 (51) | 24210 (83) | 2655 (62) | 2902 (71) |
| Yes | 2878 (49) | 4788 (17) | 1640 (38) | 1192 (29) |
| **Number of chronic conditions** | | | | |
| <4 | 3924 (67) | 24263 (84) | 3279 (76) | 3032 (74) |
| 4+ | 1924 (33) | 4735 (16) | 1016 (24) | 1062 (26) |
| **Functional impairment** | | | | |
| None | 4269 (73) | 25248 (87) | 3411 (79) | 3210 (78) |
| Mild/moderate/severe/total | 1389 (24) | 2905 (10) | 740 (17) | 767 (19) |
| **Number of depressive symptoms** | | | | |
| <10 | 377 (6) | 163 (1) | 143 (3) | 75 (2) |
| 10+ | 5471 (94) | 28834 (99) | 4152 (97) | 4019 (98) |

*(Continued)*

**Table 2.** (Continued)

| Variables | Concordant responses | | Discordant responses | |
|---|---|---|---|---|
| | Responded that they were lonely (both single item and scale) | Responded that they were not lonely (both single item and scale) | Responded that they were lonely (single item) and not lonely (scale) | Responded that they were not lonely (single item) and lonely (scale) |
| | N (%) | N (%) | N (%) | N (%) |
| **Number of social contacts** | | | | |
| High contact (4 –5) | 2392 (41) | 15489 (53) | 2149 (50) | 1783 (44) |
| Moderate contact (2–3) | 2801 (48) | 11871 (41) | 1860 (43) | 1935 (47) |
| Low contact (0–1) | 655 (11) | 1637 (6) | 286 (7) | 376 (9) |
| **Number of social activities** | | | | |
| High participation (4–5) | 912 (16) | 6536 (23) | 935 (22) | 716 (17) |
| Moderate participation (2–3) | 3279 (56) | 18158 (63) | 2588 (60) | 2410 (59) |
| Low participation (0–1) | 1647 (28) | 4282 (15) | 765 (18) | 963(24) |
| **Anxiety** | | | | |
| No | 4780 (82) | 27065 (93) | 3805 (89) | 3568 (87) |
| Yes | 1050 (18) | 1904 (7) | 476 (11) | 519 (13) |
| **Unmet need** | | | | |
| Yes | 914 (16) | 1835 (6) | 362 (8) | 504 (12) |
| No | 4917 (84) | 27122 (94) | 3927 (91) | 3577 (87) |
| **Emergency department visit** | | | | |
| Yes | 1674 (29) | 5909 (20) | 1043 (24) | 1004 (25) |
| No | 4152 (71) | 23027 (79) | 3243 (76) | 3079 (75) |
| **Care received** | | | | |
| No care received | 451 (8) | 894 (3) | 229 (5) | 236 (6) |
| Non-professional received | 352 (6) | 760 (3) | 183 (4) | 183 (4) |
| Professional received | 791 (14) | 2830 (10) | 516 (12) | 554 (14) |
| Both non-professional and professional received | 4254 (73) | 24513 (85) | 3367 (78) | 3121 (76) |

Functional impairment: assessed using the 5-point scale from the Older Americans' Resources and Services (OARS) Multidimensional Functional Assessment Questionnaire, which ranges from 2 (Excellent/Good) to 6 (Total Impairment). The CLSA modified this scale so that the range is 1–5, and the categories are (none, mild/moderate/severe/total impairment for activities of daily living). Depressive symptoms: measured using the Center for Epidemiologic Studies Depression Scale (CES-D). The 10-item CES-D scale generates a score between 0 and 30 with higher scores indicating a greater number of depressive symptoms. Social contacts: had seen any of the following social contacts within the last 6 months: children, siblings, other relatives, close friends, neighbors. Social participation: had respondents had participated in any of the following activities within the last 6 months: family or friendship activities outside the household, church or religious activities, sports or physical activities, educational and cultural activities, service club or fraternal organizational activities. For each social contact and social participation, respondents were scored from 0–5. Social contact and activity participation were categorized to the following: 0–1 = low contact/participation, 2–3 = moderate contact/participation, 4–5 = high contact/participation.

## Discussion

We compared the two measures of loneliness in the CLSA and found they demonstrated a similar prevalence of loneliness, and they comparably classified respondents as either lonely or not lonely. Respondents who were identified as lonely by both the scale and single-item were more likely to be women, less educated, and have lower income. Women tend to experience higher levels of loneliness partly due to their longer life expectancy and greater likelihood that older

**Table 3. Comparing the loneliness severity from the 3-item scale and single-item loneliness measure for the full sample and stratified by gender (Women, Men).**

**Full sample (n = 43384)**

| Single item N (%) | 3-item Loneliness Scale N (%) | | | |
|---|---|---|---|---|
| | Severely Lonely | Moderately Lonely | Not Lonely | Total |
| Severely Lonely | 2060 (5) | 1700 (4) | 824 (2) | 4618 |
| Moderately Lonely | 1722 (4) | 2578 (6) | 1259 (3) | 5587 |
| Not Lonely | 1525 (4) | 8113 (19) | 23454 (54) | 33179 |
| Total* | 5463 | 12717 | 26193 | 43384 |

**Women (n = 22094)**

| Single item N (%) | 3-item Loneliness Scale N (%) | | | |
|---|---|---|---|---|
| | Severely Lonely | Moderately Lonely | Not Lonely | Total |
| Severely Lonely | 1192 (5) | 1008 (5) | 454 (2) | 2654 |
| Moderately Lonely | 1009 (5) | 1429 (7) | 673 (3) | 3111 |
| Not Lonely | 863 (4) | 4321 (20) | 11145 (50) | 16329 |
| Total* | 3064 | 6758 | 12272 | 22094 |

**Men (n = 21091)**

| Single item N (%) | 3-item Loneliness Scale N (%) | | | |
|---|---|---|---|---|
| | Severely Lonely | Moderately Lonely | Not Lonely | Total |
| Severely Lonely | 862 (4) | 691 (3) | 369 (2) | 1922 |
| Moderately Lonely | 711 (3) | 1143 (5) | 584 (3) | 2438 |
| Not Lonely | 662 (3) | 3782 (18) | 12287 (58) | 16731 |
| Total* | 2235 | 5616 | 13240 | 21091 |

*Percentages may not sum to 100 due to rounding

women in North America live alone, which increases their risk of loneliness [26]. Lower income is also associated with loneliness, as financial constraints can limit access to social and community activities and reduce opportunities for meaningful engagement with others [26]. Similarly, individuals with lower levels of education are more likely to face socioeconomic disadvantages that restrict social participation and may contribute to social withdrawal [27]. Together, our findings are consistent with the extant literature on factors related to loneliness in older adults [26–28].

The demographic characteristics of the discordant groups were not strikingly different, aside from marital status and living arrangement (living alone). Our findings reinforce the broader recommendation outlined by the Office of National Statistics that recommends both indirect and direct measures, can provide complementary information [19]. The limited agreement we observed highlights that each measure captures somewhat different expressions or interpretations of loneliness. For this reason, we suggest that, where feasible, future CLSA users collect and examine both measures, as doing so allows them to better understand how question framing influences reported loneliness. At the same time, using both measures in subsequent analyses requires careful methodological consideration. Although collecting both is valuable, they should not be included simultaneously in statistical models due to their conceptual overlap. Instead, we recommend selecting one measure at a time, based on analytic objective. The single-item question may be more appropriate when parsimony is essential to the models. The multi-item scale may be preferable when researchers seek a more nuanced assessment of loneliness. Providing both measures allows for descriptive comparisons and subgroup exploration.

**Table 4. Descriptive characteristics of discordant respondents by loneliness severity for the 3-item scale and single-item measure.**

| | Concordant responses | | Discordant responses | |
|---|---|---|---|---|
| **Variables** | **Responded that they were not lonely (both single item and scale)** | **Responded that they were severely lonely (both single item and scale)** | **Responded that they were severely lonely (single item) and not lonely (scale)** | **Responded that they were not lonely (single item) and severely lonely (scale)** |
| | **N (%)** | **N (%)** | **N (%)** | **N (%)** |
| **Total** | 23454 (100) | 2060 (100) | 824 (100) | 1525 (100) |
| **Age** | | | | |
| 46-55 | 4266 (18) | 320 (16) | 98 (12) | 301 (20) |
| 56-65 | 7923 (34) | 701 (34) | 214 (26) | 553 (36) |
| 66-75 | 6903 (29) | 550 (27) | 241 (29) | 383 (25) |
| 76-85 | 3857 (16) | 403 (20) | 222 (27) | 238 (16) |
| 86+ | 505 (2) | 86 (4) | 49 (6) | 50 (3) |
| **Age (Mean, SD)** | 65.6 (9.9) | 66.8 (10.4) | 69.5 (10.5) | 65.2 (10.2) |
| **Gender** | | | | |
| Gender diverse | 15 (0.1) | <6 | <6 | <6 |
| Women | 11145 (48) | 1192 (58) | 454 (55) | 863 (57) |
| Men | 12287 (52) | 862 (42) | 369 (45) | 662 (43) |
| **Education** | | | | |
| Less than university | 11637 (50) | 1177 (57) | 484 (59) | 852 (56) |
| University or higher | 10696 (46) | 663 (32) | 269 (33) | 554 (36) |
| **Income** | | | | |
| <$20,000 | 538 (2) | 332 (16) | 48 (6) | 141 (9) |
| $20,000-<$50,000 | 4120 (18) | 706 (34) | 247 (30) | 428 (28) |
| $50,000+ | 17491 (75) | 823 (40) | 467 (57) | 839 (55) |
| **Marital status** | | | | |
| Single, never married | 1276 (5) | 344 (17) | 54 (7) | 214 (14) |
| Divorced/separated | 1791 (8) | 561 (27) | 90 (11) | 274 (18) |
| Married/common law | 18921 (81) | 667 (32) | 512 (62) | 864 (57) |
| Widowed | 1456 (6) | 488 (24) | 168 (20) | 172 (11) |
| **Living alone** | | | | |
| No | 20152 (86) | 929 (45) | 561 (68) | 1023 (67) |
| Yes | 3302 (14) | 1131 (55) | 263 (32) | 502 (33) |
| **Number of chronic conditions** | | | | |
| <4 | 19864 (85) | 1259 (61) | 611 (74) | 1077 (71) |
| 4+ | 3590 (15) | 801 (39) | 213 (26) | 448 (29) |
| **Functional impairment** | | | | |
| None | 20590 (88) | 1391 (68) | 632 (77) | 1148 (75) |
| Mild/moderate/severe/total | 2178 (9) | 604 (29) | 165 (20) | 332 (22) |
| **Number of depressive symptoms** | | | | |
| <10 | 106 (0.5) | 245 (12) | 45 (5) | 41 (3) |
| 10+ | 23347 (99.5) | 1815 (88) | 779 (95) | 1484 (97) |
| **Number of social contacts** | | | | |
| High contact (4–5) | 12707 (54) | 745 (36) | 409 (50) | 622 (41) |
| Moderate contact (2–3) | 9478 (40) | 1005 (49) | 346 (42) | 741 (49) |

*(Continued)*

**Table 4.** (Continued)

| Variables | Concordant responses | | Discordant responses | |
|---|---|---|---|---|
| | Responded that they were not lonely (both single item and scale) | Responded that they were severely lonely (both single item and scale) | Responded that they were severely lonely (single item) and not lonely (scale) | Responded that they were not lonely (single item) and severely lonely (scale) |
| | N (%) | N (%) | N (%) | N (%) |
| Low contact (0–1) | 1268 (5) | 310 (15) | 69 (8) | 162 (11) |
| **Number of social activities** | | | | |
| High participation (4–5) | 5379 (23) | 267 (13) | 195 (24) | 229 (15) |
| Moderate participation (2–3) | 14754 (63) | 1045 (51) | 451 (55) | 860 (56) |
| Low participation (0–1) | 3302 (14) | 743 (36) | 178 (22) | 432 (28) |
| **Anxiety** | | | | |
| No | 22018 (94) | 1584 (77) | 734 (89) | 1271 (83) |
| Yes | 1412 (6) | 468 (23) | 86 (10) | 250 (16) |
| **Unmet need** | | | | |
| Yes | 1347 (6) | 391 (19) | 68 (8) | 221 (14) |
| No | 22074 (94) | 1657 (80) | 754 (92) | 1298 (85) |
| **Emergency department visit** | | | | |
| Yes | 4635 (20) | 642 (31) | 209 (25) | 390 (26) |
| No | 18774 (80) | 1406(68) | 612 (74) | 1127(74) |
| **Care received** | | | | |
| No care received | 693 (3) | 190 (9) | 47 (6) | 103 (7) |
| Non-professional received | 561 (2) | 145 (7) | 38 (5) | 79 (5) |
| Professional received | 2158 (9) | 310 (15) | 107 (13) | 219 (14) |
| Both non-professional and professional received | 20042 (85) | 1415 (69) | 632 (77) | 1124 (74) |

Functional impairment: assessed using the 5-point scale from the Older Americans' Resources and Services (OARS) Multidimensional Functional Assessment Questionnaire, which ranges from 2 (Excellent/Good) to 6 (Total Impairment). The CLSA modified this scale so that the range is 1–5, and the categories are (none, mild/moderate/severe/total impairment for activities of daily living). Depressive symptoms: measured using the Center for Epidemiologic Studies Depression Scale (CES-D). The 10-item CES-D scale generates a score between 0 and 30 with higher scores indicating a greater number of depressive symptoms. Social contacts: had seen any of the following social contacts within the last 6 months: children, siblings, other relatives, close friends, neighbors. Social participation: had respondents had participated in any of the following activities within the last 6 months: family or friendship activities outside the household, church or religious activities, sports or physical activities, educational and cultural activities, service club or fraternal organizational activities. For each social contact and social participation, respondents were scored from 0–5. Social contact and activity participation were categorized to the following: 0–1 = low contact/participation, 2–3 = moderate contact/participation, 4–5 = high contact/participation.

Loneliness is a complex concept and one that does not have an agreed upon definition [29]. It is likely that disparate responses that we observed reflect the different ways that the questions and response categories tap into different conceptualizations of loneliness and participant comfort with attributing the term loneliness to their experience. The 3-item Loneliness Scale in the CLSA was based on the 20-item UCLA loneliness scale [11]. The developers of the 3-item scale selected the three items that reflect an individual's feeling of aloneness, rejection, and withdrawal which the developers that underlies the concept of loneliness. On the other hand, the single-item question that we examined in our study does not allude to the concepts that constitute loneliness (without ever using the word lonely), rather it poses a different challenge where it directly uses the word lonely, but then leaves what exactly loneliness means up to the interpretation of that

respondent and further asks that they ascribe that label to their experience. Cultural and social factors contribute to our understanding of the concept of loneliness and this inevitably will influence our willingness or reticence to take on the label [30,31]

In our sample, participants that self-identified as severely lonely using the single-item measure but were not considered lonely using the scale were older (69 + years of age) when we compared all concordant and discordant groups. Those that were moderately lonely in the single item but not the scale did not have common characteristics we see in lonely older adults, instead they were married, had a high income and high social contact. Criticisms of single-item, direct measures of loneliness often note that these questions will elicit socially desirable responses and that specific groups of respondents (e.g., older men) will be less likely to describe themselves as lonely. However, comparably little is known about those respondents who do willingly describe themselves as lonely but do not meet the criteria for moderate or severe loneliness in widely used scales. Our study found that these individuals have various supports, both financial and social, that some older adults do not and yet they are still lonely.

Our findings demonstrate why using two measures of loneliness can be useful, particularly for those aiming to do subsequent qualitative research to understand the context and rationale behind the responses. Although population-based surveys offer an important opportunity to examine the experience of a broad swath of older people, they cannot tease out the individual experience and how this may influence responses to single-item or multi-item scale questions. Our findings point to critical areas for future consideration for those interested in loneliness. For example, future researchers could design a sequential explanatory mixed-methods study to explore why certain respondents are classified as severely lonely on the 3-item scale but do not self-identify as lonely on the single-item question. Qualitative studies could navigate how participants interpret the wording and questions of each measure, how they understand loneliness and being lonely and what personal or cultural factors influence their willingness to express their true feeling of loneliness.

Meanwhile, the misalignment we observed in the two measures may be related to the time-based parameters in the question stems and response categories. An important component of loneliness is its temporal nature, with some research suggesting that those who are temporarily or situationally lonely (e.g., after loss of a spouse, moving to a new area) are more likely to recognize and communicate their loneliness compared to those that have been chronically lonely over a period of years [31,32]. The single-item question in the CLSA includes specific temporal boundaries, asking participants if they felt lonely over a period of days, whereas the scale does not specify a unit of time. The differences we observed in the responses to the scale versus single-item may be due to these temporal differences in the response categories. To further explore these patterns, qualitative interviews could investigate how respondents think about the "timing" of loneliness. For example, researchers could ask participants whether their feelings change or fluctuate during a day or over days or across years, and how these temporal differences shape their responses to each measure. Chronically lonely individuals have experienced loneliness over years and their loneliness can lead to an affective flattening, and an overall change in moods and attitudes which inhibits their ability to recognize their own loneliness [32]. Studies describe the ability to perceive and self-ascribe loneliness as attributable to a very specific group of lonely individuals and not all those that are labelled as lonely [31].

Overall, the discordant responses we observed are informative and highlight meaningful variability in how older adults interpret and express loneliness. These differences become clearer when the conceptual focus and temporal framing of the two measures are considered. Using both measures can therefore be valuable for researchers who are willing to explore the underlying reasons for these divergent response patterns and to gain deeper insight into how individuals understand and express their loneliness.

## Limitations

We examined data from a large population-based survey; however, as is the case in many national surveys, the respondents are relatively homogenous (e.g., White, urban-dwelling, English speaking). While we aimed to examine differences

in the measures across the spectrum of gender identities, sample size in the gender diverse group was small and made it difficult to explore differences in responses for this group. This study was cross-sectional and only included responses from baseline and the first follow-up survey. It is unclear if these comparisons will remain consistent across the further waves of data collection. Although we selected cut-offs consistent with prior research, alternative thresholds may classify respondents differently and could yield different patterns of association. We recognize that we may lose valuable information by dichotomizing scores; however, it was necessary for the purposes of our study to describe if participants were broadly identifiable as lonely or not lonely by both scale and single-item measures.

## Conclusions

Our findings suggest that the use of multiple measures of loneliness is advisable for CLSA users because they each identify critical features of respondent experience. We observed variation in respondent characteristics by comparing responses across different response categories and based on temporal periods described in question stems. We highlight future areas for research, particularly as it relates to respondent understanding of the term lonely in the single-item, direct measure and how this understanding may vary by different factors.

## Supporting information

**S1 Table. Description of the measures of loneliness in the Canadian Longitudinal Study on Aging (CLSA) follow-up 1 survey.**
(DOCX)

**S2 Table. Descriptive characteristics of CLSA Follow-up 1 survey respondents by dichotomized loneliness measure for 3-item scale and single-item.**
(DOCX)

**S3 Table. Descriptive characteristics of CLSA Follow-up 1 survey respondents by loneliness severity (single item and 3-item scale).**
(DOCX)

**S4 Table. Descriptive characteristics of discordant respondents to 3-item scale and single-item loneliness measure (moderately lonely).**
(DOCX)

## Acknowledgments

This research was made possible using the data/biospecimens collected by the Canadian Longitudinal Study on Aging (CLSA). This research has been conducted using the CLSA dataset Baseline Tracking Dataset version 3.6, Baseline Comprehensive Dataset version 4.2, Follow-up 1 Tracking version 2.1, and Follow-up 1 Comprehensive version 3.0 under Application Number 20CA004. The CLSA is led by Drs. Parminder Raina, Christina Wolfson and Susan Kirkland. The time and commitment of the participants to the CLSA study platform is gratefully acknowledged, without whom this research would not be possible.

## Author contributions

**Conceptualization:** Stephanie A Chamberlain, Lauren E. Griffith, Rachel D. Savage, Andrea Gruneir.

**Data curation:** Jesse Batara.

**Formal analysis:** Jesse Batara.

**Investigation:** Stephanie A Chamberlain.

**Methodology:** Lauren E. Griffith, Jesse Batara.

**Project administration:** Stephanie A Chamberlain.

**Resources:** Andrea Gruneir.

**Supervision:** Stephanie A Chamberlain.

**Writing – original draft:** Stephanie A Chamberlain, Andrea Gruneir.

**Writing – review & editing:** Stephanie A Chamberlain, Lauren E. Griffith, Rachel D. Savage, Jesse Batara, Wenting Yan, Andrea Gruneir.

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
