## [Decision Letter · Decision Letter 0]

17 Nov 2025

Dear Dr. Chamberlain,

Thank you for submitting your manuscript to PLOS ONE. After careful consideration, we feel that it has merit but does not fully meet PLOS ONE’s publication criteria as it currently stands. Therefore, we invite you to submit a revised version of the manuscript that addresses the points raised during the review process.

The reviewers' comments are overall positive. However, both reviewers have raised major concerns regarding conceptual and methodological aspects. The authors should address these issues thoroughly.

We look forward to receiving your revised manuscript.

Kind regards,

Rei Akaishi

Academic Editor

PLOS ONE

“: This work was supported by a Canadian Institutes of Health Research Catalyst Grant: Analysis of Canadian Longitudinal Study in Aging (CLSA) to AG (Grant #: 170313). The funder had not role in the design, conduct, interpretation, or writing of the manuscript. The opinions expressed in the manuscript are the author’s own and do not reflect the views of the Canadian Longitudinal Study on Aging.”

“This work was supported by a Canadian Institutes of Health Research Catalyst Grant: Analysis of Canadian Longitudinal Study on Aging (CLSA) to AG (Grant #: 170313). The funder had not role in the design, conduct, interpretation, or writing of the manuscript. The opinions expressed in the manuscript are the author’s own and do not reflect the views of the Canadian Longitudinal Study on Aging.”

3. For studies involving third-party data, we encourage authors to share any data specific to their analyses that they can legally distribute. PLOS recognizes, however, that authors may be using third-party data they do not have the rights to share. When third-party data cannot be publicly shared, authors must provide all information necessary for interested researchers to apply to gain access to the data. (https://journals.plos.org/plosone/s/data-availability#loc-acceptable-data-access-restrictions)

Additional Editor Comments:

The reviewers' comments are overall positive. However, both reviewers have raised major concerns regarding conceptual and methodological aspects. The authors should address these issues thoroughly.

Reviewers' comments:

Reviewer's Responses to Questions

**Comments to the Author**

1. Is the manuscript technically sound, and do the data support the conclusions?

Reviewer #1: Partly

Reviewer #2: Yes

2. Has the statistical analysis been performed appropriately and rigorously?

Reviewer #1: I Don't Know

Reviewer #2: Yes

3. Have the authors made all data underlying the findings in their manuscript fully available?

Reviewer #1: Yes

Reviewer #2: Yes

4. Is the manuscript presented in an intelligible fashion and written in standard English?

Reviewer #1: Yes

Reviewer #2: Yes

Reviewer #1: Thank you for submitting this very interesting and important piece of work. It is clear this is an important area of interest and very useful to consider these measures in tandem. I enjoyed the concept of the paper however found it at times difficult to follow so have suggestions below.

Abstract

The abstract is a bit vague in the results and discussion, more detail on e.g. the variation observed rather than just stating ‘we observed variation’ would be beneficial.

Introduction

- The introduction could benefit from deeper discussion pf previous attempts to compare across loneliness scales, even versions of the same scale e.g. 3- vs 20-item UCLA.

- It could also benefit from some mention of the psychometric literature in this space to add more depth.

Methods

- Please make the year of baseline and follow up clearer (line 102-103). At present it is unclear if the waves were both within that window of 2015 -2018 or whether baseline was 2015 and follow up was 2018.

- Lines 111-113 could be rephrased to improve readability. I would suggest ‘Our sample included participants in the baseline (2015) and first follow-up (2018) survey of the CLSA. Baseline observations were taken from the Tracking cohort (n=17,051) and Comprehensive cohort (n=27,765). Follow-up was collected using a single survey (n=44,816).’

- Line 121, make it clear they are three separate questions rather than one single question asking something like ‘do you ever feel left out, isolated, or lacking companionship?’ To someone less familiar with the UCLA scale this could be confusing.

- The word ‘so’ on line 140 us not needed grammatically.

- While it is good to refer to further detail in order work (lines 146-7) I think it would be useful to add a brief sentence summarising how social contacts and participation are defined for this study to aid the reader as otherwise this paper is unclear without reference to previous work.

- Please expand on line 167-8 how you chose that cut off. Also is there room to do sensitivity analysis with a broader cut off to reflect not lonely vs any loneliness reported, and also as sensitivity not lonely/sometimes lonely vs severe loneliness? This might reflect some of the rest of your analysis with multiple subgroups.

Results – please take a look at the readability of these results as they are quite challenging to follow at times. As you are talking about so many different cut offs across multiple scales this needs to be very clear which you are talking about when. It might be that the results would benefit from subheadings and I have added some examples of things which are unclear below.

- Please add a description of respondent characteristics to the start of the results (including a table). The demographics of the sample should be described first, particularly given for example stratification by gender is then discussed in the context of loneliness (line 200). This needs to be put into context by a description of the sample.

- Please make it clearer which measure is which in the brackets with ‘versus’ in the final paragraph on page 10. This is particularly confusing for the sentence spanning lines 187-9 but the whole paragraph could be tightened up in this respect.

- The word ‘respectively is used in line 192 however it is confusing as to whether this refers to scale or severity and doesn’t feel grammatically correct without the comparator being made clear.

- Line 194 please add a reminder of what the % was before re-classifying loneliness. Given the volume of %s in the paragraph before it, it is difficult to understand which it is comparing to. This might be helped by improved clarity on what is being compared in the ‘versus’ in the brackets as mentioned above.

- Lines 210 onwards are confusing, e.g. 21% versus 7%...does this mean that 21% of lonely people are widowed and 7% not widowed, or that 21% of widows are lonely and 7% are not lonely? This is not clear, as is the case for the other comparisons throughout that paragraph.

- Please expand on statements like ‘individuals were often classified into less severe or more severe categories depending on the scale used’ (line 240-1). E.g. who was less or more severe? Did this depend on the scale only and if so how, or was it certain characteristics and if so, can you expand on the outcomes as to which characteristics were higher or lower on each. This could be a very interesting point but needs to be expanded further.

- I wonder if the findings on lines 244-250 could be displayed graphically to aid comparison and interpretation?

- It is unclear of the timepoint where the findings are coming from given you have two waves of data. Please make this information clear rather than inferred from whether the tables are in the main paper or supplementary data. It also could be interesting to compare any prevalence or rates etc. from baseline to timepoint 1.

Discussion

- The finding described on lines 273-275 seems to be key. This could be expanded with greater discussion of how this fits with existing literature and possible reasons why this might be the case using the literature. This would make the discussion more insightful if it was discussed, and each characteristic in turn considered.

- The ONS details on lines 277-9 could benefit from being mentioned in the introduction.

- Please split the paragraph on lines 285-314 into two or three different paragraphs.

- At present it is not clear how the two waves were used or compared as findings are discussed in a more cross-sectional manner but use of terminology in the methods suggests longitudinal. It is mentioned that the data are cross sectional (line 334) however in the methods it is stated the data come from ‘baseline and the first follow-up’ (line 102). This is a bit misleading as baseline and follow up imply the data are longitudinal as people are being ‘followed up’. If it is cross-sectional please change the wording in the methods. If it is longitudinal then I would suggest that the data could be used to further explore the conclusions on lines 315-328 on chronic loneliness which are interesting.

Reviewer #2: Major Comments

The manuscript provides an interesting comparison of two possible methods for measuring loneliness in the CLSA. However, this reviewer would have liked to see a more prescriptive set of recommendations for researchers. The authors write: "[w]here time and space permits, future CLSA users should consider including both direct and indirect measures of loneliness in their analysis, as this offers an opportunity to consider how direct versus indirect questions related to loneliness may elicit different responses from study participants [pp. 2-21]". I take no issue with this advice in general, though as an applied researcher, I question whether both measures could be included as independent variables in a regression model without raising questions about multicollinearity. If loneliness is an outcome, then which measure will be more appropriate from both a conceptual and statistical point of view? Overall, I applaud the authors for their work, but they stop short of providing practical guidance that would help researchers choose the optimal variable for analysis. Readers will appreciate the addition of such prescriptive recommendations.

On p. 22, the authors state "[t]his is an example of why using two measures of loneliness can be useful, particularly for those aiming to do subsequent qualitative research to understand the context and rationale behind the responses." This point deserves some elaboration. I could see the authors conducting a sequential explanatory, mixed methods study to understand the discrepancies between the two scales. However, many researchers may simply choose one of the measures for their quantitative work and ask qualitative participants to expand upon the quantitative results from the selected measure. Perhaps the authors could solidify the practicality of their advice by providing past examples or hypothetical future examples.

Also related to practicality is the fact the direct, single-item question comes from the CES-D-10. Authors who are investigating depressive symptoms in the same study may be loathe to draw upon this question to measure loneliness because it would require breaking up the CES-D-10. With the 3-item UCLA scale available at all follow-ups after baseline, could the authors please discuss the practicality of drawing upon both measures?

Given the manuscript's topic, I was surprised to see that no comparisons included social isolation or functional social support. While social contacts and social participation form part of Menec et al.'s CLSA-based social isolation index (https://journals.plos.org/plosone/article?id=10.1371/journal.pone.0230673), neither variable captures the construct of social isolation on its own. Given the intertwined nature of loneliness, social isolation, and functional social support, I recommend the authors include these latter two variables in their comparisons.

Minor Comments

pp. 6-7: please specify that the CLSA's age- and sex-stratified random samples were stratified at the provincial level; also, please specify that the tracking cohort sample was recruited from across the 10 provinces (no geographic restriction).

p. 7: please revise the first sentence under 'study sample' to state that the participants included in your study 'completed both the baseline and follow-up 1 surveys'; the phrase "follow-up 1 survey (n = 44,816)" can be deleted for clarity.

Might the comparisons involving the direct, single-item measure and CES-D-10 be unduly correlated with one another because the direct, single-item measure is contained within the CES-D-10?

**Do you want your identity to be public for this peer review?** For information about this choice, including consent withdrawal, please see our Privacy Policy

Reviewer #1: No

Reviewer #2: No

---

## [Author Response · Author response to Decision Letter 1]

22 Dec 2025

We would like to thank the editor and reviewers for their detailed and insightful comments. We have integrated this feedback into the manuscript, and we believe that it has significantly improved the quality and clarity of the paper. Below you can find our specific responses and the location of the revisions in the manuscript.

Reviewer 1

Abstract

The abstract is a bit vague in the results and discussion, more detail on e.g. the variation observed rather than just stating ‘we observed variation’ would be beneficial.

Response: Page 3-4 (line 28-55): We have revised the abstract to include more details about our findings.

Introduction

The introduction could benefit from deeper discussion of previous attempts to compare across loneliness scales, even versions of the same scale e.g. 3- vs 20-item UCLA. It could also benefit from some mention of the psychometric literature in this space to add more depth.

Response: Page 5-6 (line 77-85): We have integrated discussion of the loneliness scales and psychometric information in the Introduction.

Methods

Please make the year of baseline and follow up clearer. At present it is unclear if the waves were both within that window of 2015 -2018 or whether baseline was 2015 and follow up was 2018.

Response: Page 7 (line 108): We have clarified the survey dates in text.

Lines 111-113 could be rephrased to improve readability. I would suggest ‘Our sample included participants in the baseline (2015) and first follow-up (2018) survey of the CLSA. Baseline observations were taken from the Tracking cohort (n=17,051) and Comprehensive cohort (n=27,765). Follow-up was collected using a single survey (n=44,816).

Response: Page 7 (line 118-122): We have revised the Methods to clarify the sample details.

Line 121, make it clear they are three separate questions rather than one single question asking something like ‘do you ever feel left out, isolated, or lacking companionship?’ To someone less familiar with the UCLA scale this could be confusing.

Response: Page 8 (line 129-133): We have revised the measures for clarity.

The word ‘so’ on line 140 us not needed grammatically.

Response: Page 9 (line 149-151): We made the revision in text.

While it is good to refer to further detail in order work (lines 146-7) I think it would be useful to add a brief sentence summarising how social contacts and participation are defined for this study to aid the reader as otherwise this paper is unclear without reference to previous work.

Response: Page 9 (line 155-167): We have provided more detail in text.

Please expand on line 167-8 how you chose that cut off. Also is there room to do sensitivity analysis with a broader cut off to reflect not lonely vs any loneliness reported, and also as sensitivity not lonely/sometimes lonely vs severe loneliness? This might reflect some of the rest of your analysis with multiple subgroups.

Response: Page 24 (line 386-388): Thank you for this suggestion. We have expanded the description of how the selected cut-offs were chosen and clarified that they were based on commonly used thresholds in the literature. We conducted additional exploratory analyses using broader classifications as an alternative to the presented groupings. These analyses produced results that were similar to those presented in the manuscript. Given this consistency and with an aim to align with established cut-offs in prior research, we elected not to include these additional results in the main text. Recognizing this, we have added to the Limitations section to acknowledge that the choice of cut-offs may influence classification and that different thresholds yield varying estimates.

Results

Please take a look at the readability of these results as they are quite challenging to follow at times. As you are talking about so many different cut offs across multiple scales this needs to be very clear which you are talking about when. It might be that the results would benefit from subheadings and I have added some examples of things which are unclear below: Please add a description of respondent characteristics to the start of the results (including a table). The demographics of the sample should be described first, particularly given for example stratification by gender is then discussed in the context of loneliness (line 200). This needs to be put into context by a description of the sample.

Please make it clearer which measure is which in the brackets with ‘versus’ in the final paragraph on page 10. This is particularly confusing for the sentence spanning lines 187-9 but the whole paragraph could be tightened up in this respect.

The word ‘respectively is used in line 192 however it is confusing as to whether this refers to scale or severity and doesn’t feel grammatically correct without the comparator being made clear.

Line 194 please add a reminder of what the % was before re-classifying loneliness. Given the volume of %s in the paragraph before it, it is difficult to understand which it is comparing to. This might be helped by improved clarity on what is being compared in the ‘versus’ in the brackets as mentioned above.

Response: Page 11-12 (line 206-211; line 213-219) Thank you for your feedback. Our full respondent demographic table is in Supplementary File 2, but we have also added a brief description in text. We have made the other requested corrections and additions in text.

Lines 210 (now it’s line 228) onwards are confusing, e.g. 21% versus 7%...does this mean that 21% of lonely people are widowed and 7% not widowed, or that 21% of widows are lonely and 7% are not lonely? This is not clear, as is the case for the other comparisons throughout that paragraph.

Response: Page 13 (line 232-236): We have revised the text for clarity.

Please expand on statements like ‘individuals were often classified into less severe or more severe categories depending on the scale used’ (line 240-1). E.g. who was less or more severe? Did this depend on the scale only and if so how, or was it certain characteristics and if so, can you expand on the outcomes as to which characteristics were higher or lower on each. This could be a very interesting point but needs to be expanded further.

Response: Page 16 (line 264-265): We have revised this in text.

I wonder if the findings on lines 244-250 could be displayed graphically to aid comparison and interpretation?

Response: Thank you for this suggestion. We explored options for presenting these comparisons, as we recognize that the tables are extensive. However, because the variables included in these analyses span different domains and involve a mix of categorical and ordinal data, a single figure or group of figured became visually dense and difficult to interpret. Creating a series of separate figures resulted in fragmentation and reduced visual comparability across groups. We determined that the tabular format, currently presented in the Supplementary Files, remains the most interpretable way to display these results.

It is unclear of the timepoint where the findings are coming from given you have two waves of data. Please make this information clear rather than inferred from whether the tables are in the main paper or supplementary data. It also could be interesting to compare any prevalence or rates etc. from baseline to timepoint 1.

Response: Page 7 (line 118-122): We agree that comparing data across time points would be valuable. However, the 3-item Loneliness Scale was only first included in the first follow-up wave, so we were unable to make this comparison in the baseline data. This has been added to the manuscript.

Discussion

The finding described on lines 273-275 seems to be key. This could be expanded with greater discussion of how this fits with existing literature and possible reasons why this might be the case using the literature. This would make the discussion more insightful if it was discussed, and each characteristic in turn considered.

Response: Page 20 (line 298-304): We have revised the Discussion based on your suggestion.

The ONS details on lines 277-9 could benefit from being mentioned in the introduction.

Response: Page 2 (line 95-97): Thank you for this suggestion. We have revised the Introduction.

Please split the paragraph on lines 285-314 into two or three different paragraphs.

Response: Page 21-23 (line 321-357): Thank you for this suggestion, we have split this part into paragraphs.

At present it is not clear how the two waves were used or compared as findings are discussed in a more cross-sectional manner but use of terminology in the methods suggests longitudinal. It is mentioned that the data are cross sectional (line 334) however in the methods it is stated the data come from ‘baseline and the first follow-up’ (line 102). This is a bit misleading as baseline and follow up imply the data are longitudinal as people are being ‘followed up’. If it is cross-sectional please change the wording in the methods. If it is longitudinal then I would suggest that the data could be used to further explore the conclusions on lines 315-328 on chronic loneliness which are interesting.

Response: Page 7 (line 118-122): Thank you for highlighting this point. We agree the use of a longitudinal survey may create the impression that the analysis was longitudinal when it was not. To clarify, although our sample was drawn from participants who completed both the baseline and follow-up 1 waves of the CLSA, loneliness was only measured at follow-up 1. As a result, the analysis is cross-sectional, and baseline data were used to define the sample and not to examine change over time. We have revised the Methods to make this more explicit.

Reviewer 2

The manuscript provides an interesting comparison of two possible methods for measuring loneliness in the CLSA. However, this reviewer would have liked to see a more prescriptive set of recommendations for researchers. The authors write: "[w]here time and space permits, future CLSA users should consider including both direct and indirect measures of loneliness in their analysis, as this offers an opportunity to consider how direct versus indirect questions related to loneliness may elicit different responses from study participants [pp. 2-21]". I take no issue with this advice in general, though as an applied researcher, I question whether both measures could be included as independent variables in a regression model without raising questions about multicollinearity. If loneliness is an outcome, then which measure will be more appropriate from both a conceptual and statistical point of view? Overall, I applaud the authors for their work, but they stop short of providing practical guidance that would help researchers choose the optimal variable for analysis. Readers will appreciate the addition of such prescriptive recommendations.

Response: Page 19 (line 308-320):Thank you for this thoughtful comment. We agree that clearer, more prescriptive guidance could strengthen the manuscript and we have revised the Discussion accordingly. In this addition, we clarify that the two measures should not be used simultaneously due to conceptual overall and risk of multicollinearity.

On p. 22, the authors state "[t]his is an example of why using two measures of loneliness can be useful, particularly for those aiming to do subsequent qualitative research to understand the context and rationale behind the responses." This point deserves some elaboration. I could see the authors conducting a sequential explanatory, mixed methods study to understand the discrepancies between the two scales. However, many researchers may simply choose one of the measures for their quantitative work and ask qualitative participants to expand upon the quantitative results from the selected measure. Perhaps the authors could solidify the practicality of their advice by providing past examples or hypothetical future examples.

Response: Page 22-23 (line 352-357; line 366-370): Thank you for your suggestion. We have revised the Discussion accordingly.

Also related to practicality is the fact the direct, single-item question comes from the CES-D-10. Authors who are investigating depressive symptoms in the same study may be loathe to draw upon this question to measure loneliness because it would require breaking up the CES-D-10. With the 3-item UCLA scale available at all follow-ups after baseline, could the authors please discuss the practicality of drawing upon both measures?

Response: Page 24 (line 375-380): Thank you for your comment. We agree that breaking up the CES-D-10 and just using the loneliness item would disrupt the CES-D-10’s psychometric integrity, but we still think the presence of the single-item question in the CLSA remains analytically valuable based on the discrepancy we observed in this study. We have expanded on this in our Discussion.

Given the manuscript's topic, I was surprised to see that no comparisons included social isolation or functional social support. While social contacts and social participation form part of Menec et al.'s CLSA-based social isolation index (https://journals.plos.org/plosone/article?id=10.1371/journal.pone.0230673), neither variable captures the construct of social isolation on its own. Given the intertwined nature of loneliness, social isolation, and functional social support, I recommend the authors include these latter two variables in their comparisons.

Response: We appreciate the importance and relevance of social isolation to our understanding of loneliness. In our study, we focused on descriptive comparisons of individual components that typically comprise social isolation indices in the CLSA, including living alone, marital status, social participation, and social contacts. Our aim was to explore how these individual aspects of isolation relate to the concordance and discordance classifications of loneliness, rather than create or rely on a composite social isolation index. By including these variables separately, we can provide a more nuanced description of specific factors that are associated with loneliness. We have clarified this rationale in the Methods.

pp. 6-7: please specify that the CLSA's age- and sex-stratified random samples were stratified at the provincial level; also, please specify that the tracking cohort sample was recruited from across the 10 provinces (no geographic restriction).

Response: Page 5 (line 111-113): We have revised the Methods accordingly.

please revise the first sentence under 'study sample' to state that the participants included in your study 'completed both the baseline and follow-up 1 surveys'; the phrase "follow-up 1 survey (n = 44,816)" can be deleted for clarity.

Response: Page 7 (line 118-120): We have made this revision in text.

Might the comparisons involving the direct, single-item measure and CES-D-10 be unduly correlated with one another because the direct, single-item measure is contained within the CES-D-10?

Response: We wish to clarify that our analyses did not compare the single-item question with other individual items in the CESD, nor did we analyze the overall CESD score. It was used independently and not combined or contrasted against the broader depression scale.

---

## [Decision Letter · Decision Letter 1]

8 Jan 2026

Single-item versus scale: Comparing respondent demographic, social, and health characteristics by measure of loneliness using the Canadian Longitudinal Study on Aging (CLSA) data

PONE-D-25-41070R1

Dear Dr. Chamberlain,

We’re pleased to inform you that your manuscript has been judged scientifically suitable for publication and will be formally accepted for publication once it meets all outstanding technical requirements.

Kind regards,

Rei Akaishi

Academic Editor

PLOS One

Additional Editor Comments (optional):

The authors addressed concerns of the reviewers raised in the previous round. But please address the additional minor points raised by Reviewer 2.

Reviewers' comments:

Reviewer's Responses to Questions

**Comments to the Author**

Reviewer #1: All comments have been addressed

Reviewer #2: All comments have been addressed

2. Is the manuscript technically sound, and do the data support the conclusions?

Reviewer #1: Yes

Reviewer #2: Yes

3. Has the statistical analysis been performed appropriately and rigorously?

Reviewer #1: Yes

Reviewer #2: Yes

4. Have the authors made all data underlying the findings in their manuscript fully available?

Reviewer #1: Yes

Reviewer #2: Yes

5. Is the manuscript presented in an intelligible fashion and written in standard English?

Reviewer #1: Yes

Reviewer #2: Yes

Reviewer #1: (No Response)

Reviewer #2: The authors have satisfactorily addressed all my comments. I just have a few minor issues for the authors to consider.

P. 3, line 36: the years 2015-2018 appear to apply to baseline and follow-up 1, when in fact they do not apply to baseline.

P. 7, line 109: the phrasing could more clearly identify that the 3-items Loneliness Scale was introduced at follow-up 1.

P. 7, lines 118-120: the text could more clearly indicate that the numbers of participants come from those who completed both the baseline and follow-up 1 surveys. Therefore, to be in this study, participants had to complete the baseline and follow-up 1 surveys.

P. 8, line 140: the CESD10 score range is 0 to 3.

P. 9, lines 161-162: clarify whether the numbers, eg, 4-5, are numbers of contacts. If number of contacts, then did no participant report more than 5 contacts.

P. 9, line 163: use US spelling in ‘categorization’.

P. 11, line 196: ‘we’ instead of ‘were’?

P. 13, lines 233-234: the phrase beginning with ‘and’ is not a complete sentence.

P. 24, line 390: ‘measure’ should be plural.

**Do you want your identity to be public for this peer review?** For information about this choice, including consent withdrawal, please see our Privacy Policy

Reviewer #1: No

Reviewer #2: No

---

## [Editor Report · Acceptance letter]

PONE-D-25-41070R1

PLOS One

Dear Dr. Chamberlain,

I'm pleased to inform you that your manuscript has been deemed suitable for publication in PLOS One. Congratulations! Your manuscript is now being handed over to our production team.

Kind regards,

on behalf of

Dr. Rei Akaishi

Academic Editor

PLOS One